# Anti-Inflammatory and Immune Modulatory Effects of Synbio-Glucan in an Atopic Dermatitis Mouse Model

**DOI:** 10.3390/nu13041090

**Published:** 2021-03-26

**Authors:** Yoon-Hwan Kim, Min Soo Kang, Tae Hyeong Kim, Yunho Jeong, Jin-Ok Ahn, Jung Hoon Choi, Jin-Young Chung

**Affiliations:** 1Department of Veterinary Internal Medicine and Institute of Veterinary Science, College of Veterinary Medicine, Kangwon National University, Chuncheon-si 24341, Korea; luperskim@gmail.com (Y.-H.K.); jsteve35@outlook.com (Y.J.); joahn@kangwon.ac.kr (J.-O.A.); 2Department of Veterinary Anatomy and Institute of Veterinary Science, College of Veterinary Medicine, Kangwon National University, Chuncheon-si 24341, Korea; imkangms@kangwon.ac.kr (M.S.K.); xogudsla9402@kangwon.ac.kr (T.H.K.)

**Keywords:** atopic dermatitis, β‐glucan, house dust mite, Nc/Nga mice, probiotics

## Abstract

Many trials have been conducted to treat atopic dermatitis (AD), but these therapies are generally unsuccessful because of their insufficiency or side effects. This study examined the efficacy of β-glucan derived from oats with fermented probiotics (called Synbio-glucan) on an AD-induced mouse model. For the experiment, Nc/Nga mice were exposed to a house dust mite extract (HDM) to induce AD. The mice were placed in one of four groups: positive control group, Synbio-glucan topical treatment group, Synbio-glucan dietary treatment group, and Synbio-glucan topical + dietary treatment group. The experiment revealed no significant difference in the serum IgE concentration among the groups. Serum cytokine antibody arrays showed that genes related to the immune response were enriched. A significant difference in the skin lesion scores was observed between the groups. Compared to the control group tissue, skin lesions were alleviated in the Synbio-glucan topical treatment group and Synbio-glucan dietary treatment group. Interestingly, almost normal structures were observed within the skin lesions in the Synbio-glucan topical + dietary treatment group. Overall, the β-glucan extracted from oats and fermented probiotic mixture is effective in treating atopic dermatitis.

## 1. Introduction

Atopic dermatitis (AD) is a common chronic inflammatory skin disease and a global public health concern because of its increasing prevalence and socioeconomic burden [1]. Approximately 20% of people around the world suffer from AD [2]. The onset age of AD varies from infants to adults and one study showed that adult-onset AD occurs in 36.8% of cases. Approximately 47.6% of adult patients with AD showed a persistent and chronic pattern of AD. The most common AD phenotype in adult patients is lichenified/exudative flexural dermatitis (48.5%); eczema and prurigo can also occur [3].

Many trials have been conducted to overcome AD, including antihistamines, glucocorticoids, and anti-microbial agents. More advanced therapies, such as immunotherapy, have also been used to treat AD. On the other hand, these therapies have been unsuccessful because of their insufficiency or side effects [4]. Various new drugs that are specific monoclonal antibodies and new topical molecules that are expected to block one of the mechanisms of AD, such as Tofacitinib, Crisaborole, and Dupilumab, have been developed. These new drugs have introduced a new era in the treatment of AD [5]. On the other hand, their results in treating AD have been unsatisfactory. Therefore, recent studies have focused on alternative medicine to control AD [6,7,8].

β-glucan is a biologically activated polysaccharide found in the cell walls of algae, cereals, mushrooms, yeast, and some bacteria [9]. This polysaccharide has potent immunomodulatory effects on the innate and adaptive immunity. Anti-cancer effects against the proliferation of tumor cells and of the promotion of interleukins (ILs) have also been reported [10,11]. β-glucan is also involved in cardiovascular disease, hyperlipidemia, and hypercholesterolemia [12].

Several studies have applied β-glucan to a range of allergic diseases [13], but its efficacy against AD is controversial. One study reported that the oral administration of β-glucan derived from yeast (*Aureobasidium pullulans*) has anti-AD effects in animal models [14]. Another study indicated anti-AD effects through the topical administration of β-glucan derived from mushroom, *Pleurotus ostreatus*, on human patients [15].

Probiotics have attracted interest as an alternative medicine to control AD. Several studies have examined the efficacy of probiotics on AD, but the results have been controversial [16]. One meta-analysis confirmed that probiotics administered prenatally and postnatally could effectively reduce the risk of atopy [17].

This study examined the efficacy of β-glucan derived from oats with probiotics in an AD-induced mouse model.

## 2. Materials and Methods

### 2.1. Animals

Twenty female Nishiki-nezumi Cinnamon/Nagoya (Nc/Nga, 8-week-old) mice, for the AD-developing mouse model, were purchased from Central Laboratory Animal Inc. (Seoul, Korea). The experimental protocols complied with the ethical guidelines, with approval obtained from the Kangwon National University Institutional Care and Animal Use Committee (KW-190521-2). The mice were housed in an air-conditioned conventional room maintained at 24 ± 2 °C and 55 ± 15% humidity.

The mice were divided into four groups: positive control (house dust mite extract (HDM) treatment only), Synbio-glucan topical treatment (Synbio-glucan topical treatment in HDM-treated mice), Synbio-glucan dietary treatment (Synbio-glucan dietary treatment in HDM-treated mice), and Synbio-glucan topical + dietary treatment groups (Synbio-glucan topical + dietary treatment in HDM-treated mice) (*n* = 5 in each group).

### 2.2. Drugs and Reagents

Synbio-glucan is composed of β-glucan, avenanthramides, oat lipids, oat peptides, oat flavonoids (phenolic structure), tocopherol (Vit. E), and sphingomyelinase (patent number:10-1805863). For the production of Synbio-glucan, the beans and oats were preprocessed by heating over 80 °C. Subsequently, they were fermented with probiotics (*Lactobacillus plantarum*, *Bifidobacterium longum*, and *Pediococcus pentosaceus*).

The Synbio-glucan diet was composed of a standard diet (including an 18% protein rodent diet), containing 2% Synbio-glucan agent (Koatech, KyoungGido, Korea). A HDM allergen ointment composed of *Dermatophagoides farinae* was purchased from Biostir Inc. (Kobe, Japan).

### 2.3. Induction of AD 

AD was induced by treatment with 100 µL of 4% (*w*/*v*) sodium dodecyl sulfate (SDS; Sigma-Aldrich, St. Louis, MO, USA) after shaving hair on the back to disrupt the skin barrier. After drying SDS, 100 mg of HDM allergen (HDM, Biostir Inc., Kobe, Japan) per mouse was applied on the bared skin region twice weekly for 4 weeks (Figure 1).

### 2.4. Treatment of Synbio-Glucan in the NC/Nga Mice 

For the Synbio-glucan topical treatment group, the HDM-applied skin area was treated with 100 µL of Synbio-glucan every day for three weeks, from one week after the HDM treatment. After drying the Synbio-glucan topically treated skin, the mice were returned to their inhabited cases. All mice of this group were provided with a standard diet. For the Synbio-glucan dietary treatment group, the Synbio-glucan diet was provided for four weeks from the start of the experiment. For the positive control group (HDM treatment only), 100 µL PBS was used instead of the Synbio-glucan topical agent for three weeks from one week after the HDM treatment. All the mice in this group were provided with a standard diet. The Synbio-glucan topical and dietary treatment group was given the Synbio-glucan diet for four weeks from the start of the experiment. The HDM-applied skin area was treated with 100 µL of Synbio-glucan every day for three weeks, starting from one week after the HDM treatment (Figure 1).

### 2.5. Serum IgE Concentration Assay

The serum was collected from sacrificed mice. The total serum IgE concentration was measured using an ELISA kit (Fujifilm Wako Shibayagi Corporation, Shibukawa, Japan), following the manufacturer’s instructions. The plate was analyzed by a SpectraMax ABS Plus Microplate Reader (Molecular Devices, LLC, San Jose, CA, USA) at 450 nm.

### 2.6. Serum Cytokine Antibody Assay

The serum (50 µL) obtained from the sacrificed mice was used for the array protocol. The relative serum cytokine levels were measured by mouse cytokine antibody array L308 membrane kit (RayBiotech, Inc., Norcross, GA, USA), and 100 μL of serum pooled from five mice per group was used for the cytokine array. The diluted pooled serum (1:10) was probed following the manufacturer’s protocols to determine the cytokine profile. The fold changes of cytokine were calculated as the relative values of the treated groups corresponding to that of the control group.

### 2.7. Scoring of Skin Lesions

The extent of erythema/hemorrhage, scarring/dryness, edema, and excoriation/erosion was scored individually as 0 (none), 1 (mild), 2 (moderate), and 3 (severe). The total skin score was the sum of the individual scores [18,19] (Appendix A). Scoring was performed every week during the experimental period.

### 2.8. Histological Analysis

The mice were perfused transcardially using 0.1 M phosphate-buffered saline (PBS) after deep anesthesia by a high dose of Zoletil 50^®^ (Virbac, Carros, France) at the end of the experiments. The fixation was performed using 4% paraformaldehyde in 0.1 M PBS subsequently. The skin was collected and fixed with same fixation at 4 °C for 24 h. The tissues were embedded with paraffin after dehydration processes. The embedded tissues were cut into 5-μm-thick sections using a microtome (Leica Microsystems GmbH, Wetzlar, Germany). The tissues were mounted on slides (Muto Pure Chemicals Co., Ltd., Tokyo, Japan) and stained with hematoxylin and eosin (H&E) and toluidine blue (TB) using the standard protocol.

### 2.9. Statistical Anlysis

The data were analyzed by statistical analysis software (GraphPad Prism, Ver. 5.01, San Diego, CA, USA). The represented values are the means of the experiments of each group. The differences among the means were identified using Mann–Whitney and Kruskal–Wallis tests. Statistical significance was considered as a *p*-value < 0.05.

## 3. Results

### 3.1. Comparison of the Serum IgE Concentration between Groups 

Atopic dermatitis was induced in Nc/Nga mice. Such mice were treated with Symbio-glucan topically, dietary, dietary plus topically, or left untreated. Normally, atopic dermatitis tends to produce an excessive IgE level. The serum IgE concentration in the groups was compared by collecting the serum of all the groups from sacrificed mice. The serum IgE concentrations of the mice in all the groups were similar (*p* = 0.2560). The serum IgE concentration was similar in the positive control group, Synbio-glucan topical treatment group (*p* = 0.5476), Synbio-glucan dietary treatment group (*p* = 0.3095), and Synbio-glucan topical + dietary treatment group (*p* = 0.3095). The serum IgE concentrations in the Synbio-glucan dietary treatment group (*p* = 0.3095), Synbio-glucan topical + dietary treatment group (*p* = 0.3095), and Synbio-glucan topical treatment group were similar. The serum IgE concentrations in the Synbio-glucan dietary treatment group and Synbio-glucan topical + dietary treatment group were similar (*p* = 0.1508) (Figure 2).

### 3.2. Comparison of the Serum Cytokine Antibody Arrays between Groups

Figure 3A–C present the antibody array scatter plots. The scatter plots represent the fold changes in the serum cytokines between the treatment groups and positive control group. The plots revealed changes in the signal intensities between the Synbio-glucan topical treatment group and the positive control group (Figure 3A), the Synbio-glucan dietary treatment group and the positive control group (Figure 3B), as well as between the Synbio-glucan topical + dietary treatment group and the positive control group (Figure 3C). The red and green lines indicate two-fold up- or downregulated expression, respectively. The data indicated by the red and green dots over the red and green lines are presented in the tables. This result confirmed the patterns of changes in the serum cytokines between the treatment groups and positive control group.

A comparison of the serum cytokine array of each treatment group with the positive control group identified the up- or downregulated proteins. In the Synbio-glucan topical treatment group, 27 proteins were significantly downregulated (>2-fold changes in the normalized value; *t*-test *p*-value < 0.05; Table 1). In the Synbio-glucan dietary treatment group, there were 27 significantly upregulated proteins (>2-fold changes in the normalized value; *t*-test *p*-value < 0.05; Table 2). In the Synbio-glucan topical + dietary treatment group, 45 upregulated proteins and 12 downregulated proteins were confirmed (>2-fold changes in the normalized value; *t*-test *p*-value < 0.05; Table 3).

All proteins were identified from the antibody array analysis and were analyzed further according to the categories within The Database for Annotation, Visualization and Integrated Discovery (DAVID) and the Kyoto Encyclopedia of Genes and Genomes (KEGG). DAVID is a database resource for analyzing the biological gene functions. KEGG is a database resource for analyzing the pathways related to biological systems. Fifty-five components were enriched significantly in the Gene Ontology_Biological Process (GO_BP) by the DAVID results for the Synbio-glucan topical treatment group. Among them, the top 10 enriched GO_BP terms were the response to lipopolysaccharide, inflammatory response, immune response, response to glucocorticoid, innate immune response, regulation of cell proliferation, wound healing, response to ethanol, response to the drug, and immune system process (Figure 3D). Among the KEGG categories, 23 pathways were enriched significantly in the Synbio-glucan topical treatment group. The top five enriched KEGG categories were the cytokine–cytokine receptor interaction, PI3K-Akt signaling pathway, Chagas disease (American trypanosomiasis), TNF signaling pathway, and osteoclast differentiation (Figure 3D).

As determined by applying the GO_BP, fifty-eight components in the list of proteins regulated in the Synbio-glucan dietary treatment group were enriched significantly. Among the proteins, the top 10 enriched GO_BP terms were the immune responses, chemokine-mediated signaling pathway, inflammatory response, cell–cell signaling, positive regulation of phosphatidylinositol 3-kinase signaling, positive regulation of ERK1 and ERK2 cascade, cell chemotaxis, chemotaxis, lymphocyte chemotaxis, and protein kinase B signaling (Figure 3E). Among the KEGG categories, seven pathways were enriched significantly in the Synbio-glucan dietary treatment group. Among the pathways, the top five enriched KEGG categories were the cytokine–cytokine receptor interaction, Jak-STAT signaling pathway, chemokine signaling pathway, rheumatoid arthritis, and NF-kappa B signaling pathway (Figure 3E).

In the GO_BP results for the Synbio-glucan topical + dietary treatment group, 109 components were enriched significantly among the list of proteins. Among them, the top 10 enriched GO_BP terms were the immune responses, positive regulation of peptidyl-tyrosine phosphorylation, chemotaxis, chemokine-mediated signaling pathway, positive regulation of cell proliferation, inflammatory response, positive regulation of inflammatory response, cell chemotaxis, negative regulation of cell proliferation, and negative regulation of viral genome replication (Figure 3F). Among the KEGG results, 12 pathways were enriched significantly. The top five enriched pathways were the cytokine–cytokine receptor interaction, Jak-STAT signaling pathway, chemokine signaling pathway, hematopoietic cell lineage, and PI3K-Akt signaling pathway (Figure 3F).

### 3.3. Comparison of Skin Lesion Scores between Groups

The skin lesions included erythema/hemorrhage, scarring/dryness, edema, and excoriation/erosion (Figure 4A). These clinical signs were more severe in the positive control group and Synbio-glucan dietary treatment group than in the other groups. Compared to the positive control group, the scores of the lesions in the Synbio-glucan topical treatment group (*p* = 0.0432) and Synbio-glucan topical + dietary treatment group (*p* = 0.0273) showed significant differences, but not in the Synbio-glucan dietary treatment group (*p* = 0.8294). The skin lesion scores were similar in the Synbio-glucan topical treatment group, the Synbio-glucan dietary treatment group (*p* = 0.0532), and Synbio-glucan topical + dietary treatment group (*p* = 0.7449). The scores of the skin lesions were significantly different in Synbio-glucan dietary treatment group and Synbio-glucan topical + dietary treatment group (*p* = 0.0345) (Figure 4B).

### 3.4. Comparison of Histological Results between the Groups

H&E staining of the positive control group’s tissue revealed epidermal and dermal hyperplasia, excessive keratinization, and infiltration of lymphocytes. Compared to the positive control group, the tissue, epidermal and dermal hyperplasia, keratinization, and infiltration of lymphocytes were alleviated in the other groups. The Synbio-glucan topical treatment group showed greater alleviation of epidermal and dermal hyperplasia, keratinization, and infiltration of lymphocytes than the Synbio-glucan dietary treatment group. Interestingly, almost normal structures were observed within the epidermis, dermis, subcutaneous layer, and muscle layer in the Synbio-glucan topical + dietary treatment group (Figure 5A).

The TB staining results showed that the number of mast cells in the dermis was prominent in the positive control group. The number of mast cells decreased in the Synbio-glucan topical treatment group and Synbio-glucan dietary treatment group. Mast cells were decreased more in the Synbio-glucan topical treatment group than the Synbio-glucan dietary treatment group. The most prominent mast cells decreased in the Synbio-glucan topical + dietary treatment group (Figure 5B).

## 4. Discussion

Synbio-glucan (β-glucan extracted from oats and probiotic mixture) was applied to an AD-induced Nc/Nga mouse model with HDM. β-glucan is one of the structural components of the cell walls of bacteria, algae, fungi, yeasts, and cereals [20]. In cereals (oats and wheat), β-glucan exists as linear polysaccharides, in which glucose monomers are bound by β-(1,3) and β-(1,4) linkages [21]. β-glucan is a water-soluble fiber. Many studies reported that β-glucan has a positive influence on the physiological and metabolic processes in the body. Previous studies confirmed that this type of β-glucan prevents obesity and reduces blood glucose, cholesterol concentrations, and body weight [22,23]. Some studies used β-glucan to treat or prevent atopic dermatitis in vivo, but the β-glucan used was derived from algae and yeast not from cereals, such as oats [14,15,24].

In this study, β-glucan derived from oats was fermented with probiotics composed of *Lactobacillus plantarum*, *Bifidobacterium longum,* and *Pediococcus pentosaceus* to produce Synbio-glucan. Probiotics have a health-promoting effect. Among the several strains of probiotics, *Lactobacillus plantarum* is commonly found in many fermented food products and has been used to regulate the immune system [25]. A few trials to treat atopic dermatitis with *Lactobacillus plantarum* have been conducted. These studies confirmed that *Lactobacillus plantarum* alleviated AD [26,27]. *Bifidobacterium* is a normal bacterium in the intestines of animals and is used widely as an immune regulatory supplement in the intestines. A representative probiotic, *Bifidobacterium longum*, has been studied as an immunomodulatory agent in atopic dermatitis [28,29]. *Pediococcus pentosaceus* is categorized as a “lactic acid bacterium”, and the related research on *Pediococcus pentosaceus* has focused on food preservation [30]. Only a few studies reported its preventive effects in a food allergy model, but there are no reports on its effects on atopic dermatitis [31].

The skin lesion scores and histological analysis showed that the Synbio-glucan topical + dietary treatment markedly alleviated the skin lesions. Both the Synbio-glucan topical treatment and the Synbio-glucan dietary treatment also improved the skin lesions. On the other hand, the improvement of the skin lesion in the Synbio-glucan dietary treatment was significantly lower. Previous studies confirmed that the oral administration of β-glucan and probiotics alleviated AD significantly [14,27]. In the present study, the scores of skin lesions were similar to those of the positive control, but histological analysis showed improvement of the skin lesion compared to the control group. This discrepancy in the results was attributed to the percentage of β-glucan and probiotics dietary supplements. Therefore, further study will be needed to elucidate this discrepancy of the dietary effects using a high percentage of β-glucan derived from oats. Nevertheless, the present results showed that topical application with dietary administration is more effective on AD-induced skin lesions than dietary administration only. AD is a two-phase chronic inflammatory skin disease accompanied by erythema, edema, excoriation, or lichenification. AD features a Th2-type disease, with the infiltration of various immune cells stimulating B-cells in the initial phase. In the later phase, the Th1 cytokines effectively promote the cellular immune response [32,33]. Of an antibody array with 308 cytokines in the Synbio-glucan topical treatment group, only 27 cytokines were downregulated more than two-fold. The least downregulated one was IFN-β, which exerts antiviral, antiproliferative, and immunomodulatory activities. In immune cells, IFN-β is produced naturally in the skin by dermal dendritic cells under biological or chemical stimulation [34]. The downregulated IFN-β in this study might be related to a decrease in the number of dendritic cells and their immune regulatory function. On the other hand, few studies of IFN-β with AD have been reported. Therefore, more research will be needed to prove the correlation between IFN-β and AD.

Many studies have reported high levels of IL-9 expression in AD [35,36]. Interestingly, despite the alleviation of the lesions in the topical and dietary treatment groups, IL-9 was expressed strongly in the dietary treatment group and topical + dietary treatment group, except for the topical treatment group. The meaning of this result is unclear. One study suggested that β-glucan might encourage allergic inflammation under certain conditions. On the other hand, this group could not determine the clinical significance of their observations [37].

The type of cytokine and the degree of expression were confirmed in each treatment group using a serum cytokine antibody array. Moreover, the activated biological processes and pathways could be identified. The entirety of the cytokine expression and biological processes and pathways when β-glucan is applied to an AD-induced mouse model is unclear. GO is an ontology used widely in bioinformatics for annotating large-scale genes and gene products [38]. KEGG is a practical database resource for genome sequencing and polymer experiment technology [39]. In this study, the biological gene functions and the pathway related to the biological system were analyzed using GO and KEGG.

## 5. Conclusions

This study examined the efficacy of β-glucan derived from oats with probiotics in an AD-induced mouse model. The results confirmed that Synbio-glucan could improve lesions in an AD-induced mouse model with HDM. To the best of the authors’ knowledge, this is the first study to evaluate the effects of β-glucan extracted from oats and a probiotic mixture to treat atopic dermatitis.

## Figures and Tables

**Figure 1 nutrients-13-01090-f001:**
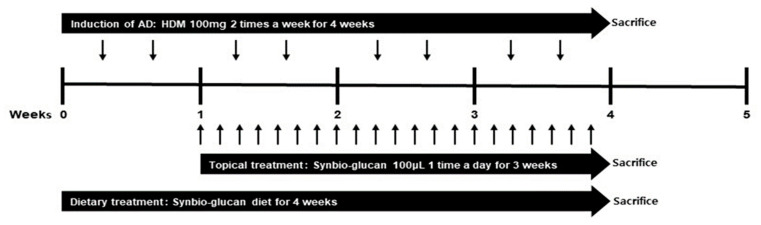
Experimental scheme.

**Figure 2 nutrients-13-01090-f002:**
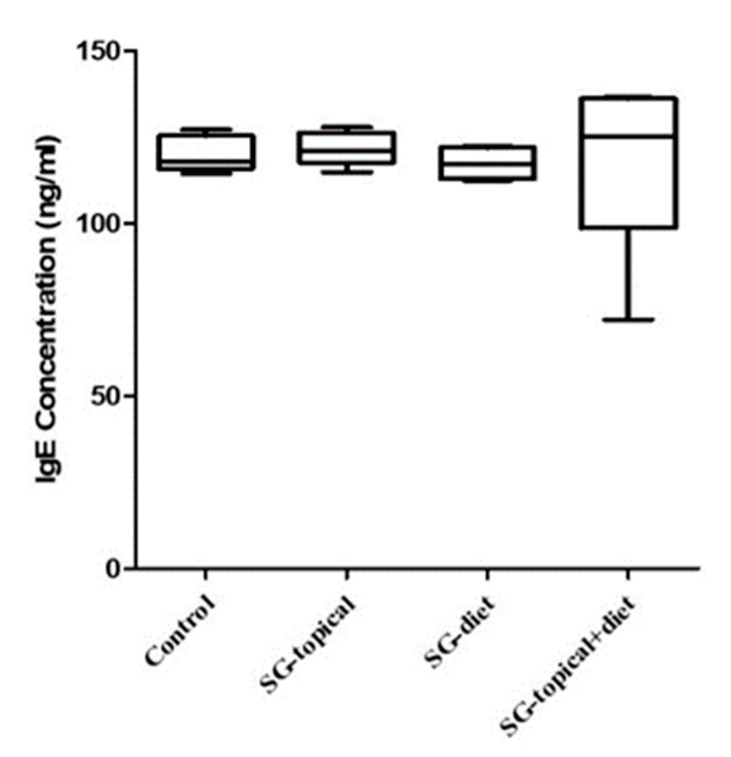
Comparison of the serum IgE concentrations between the groups. The serum IgE concentrations are expressed as the optic density units. The serum IgE concentrations were similar in all the groups. Control; positive control group, SG-topical; Synbio-glucan topical treatment group, SG-diet; Synbio-glucan dietary treatment group, SG-topical + diet; Synbio-glucan topical + dietary treatment group.

**Figure 3 nutrients-13-01090-f003:**
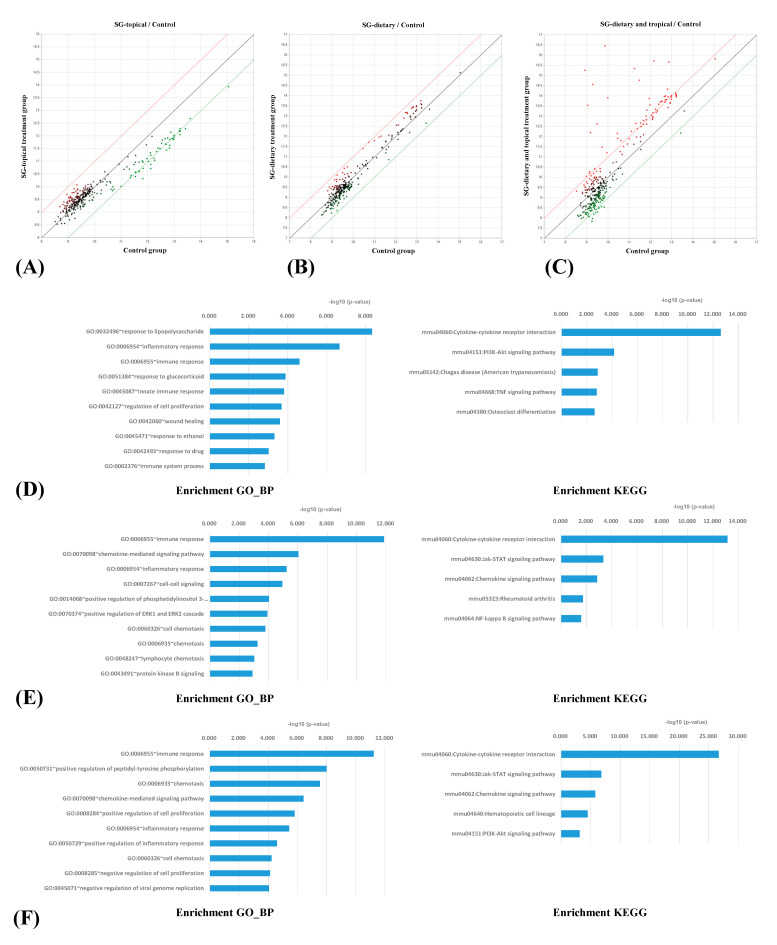
(**A**) Changes in signal between the Synbio-glucan topical treatment group and the positive control group, (**B**) Synbio-glucan dietary treatment group and the positive control group, and between (**C**) the Synbio-glucan topical + dietary treatment group and the positive control group. The red and green lines indicate two-fold up- or downregulated expression, respectively. This scatter plot shows the pattern of changes in the serum cytokines between the treatment groups and the positive control group. (**D**) Functional analysis of the antibody array results after the Synbio-glucan topical treatment. The most enriched GO_BP term was the response to lipopolysaccharide and the most enriched KEGG categories were the cytokine–cytokine receptor interactions. (**E**) Functional analysis of the antibody array results after the Synbio-glucan dietary treatment. The most enriched GO_BP term was the immune responses and the most enriched KEGG categories were the cytokine–cytokine receptor interactions. (**F**) Functional analysis of the antibody array results after the Synbio-glucan topical + dietary treatment. The most enriched GO_BP term was the immune responses, and the most enriched KEGG categories were the cytokine–cytokine receptor interactions.

**Figure 4 nutrients-13-01090-f004:**
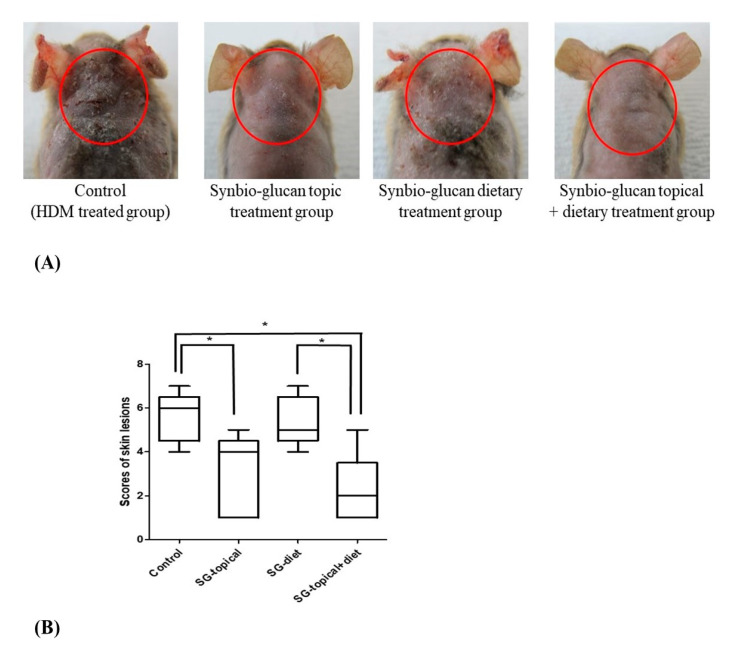
Scores of the skin lesions. (**A**) Representative clinical symptoms in the positive control group, Synbio-glucan topical treatment group, Synbio-glucan dietary treatment group, and Synbio-glucan topical + dietary treatment group. (**B**) Scores of the skin lesions of all the groups were significantly different, except for the scores between the positive control group and Synbio-glucan dietary treatment group, between the Synbio-glucan topical treatment group and Synbio-glucan dietary treatment group, and between the Synbio-glucan topical treatment group and Synbio-glucan topical + dietary treatment group. Control; positive control group, SG-topical; Synbio-glucan topical treatment group, SG-diet; Synbio-glucan dietary treatment group, SG-topical + diet; Synbio-glucan topical + dietary treatment group. * *p* < 0.05.

**Figure 5 nutrients-13-01090-f005:**
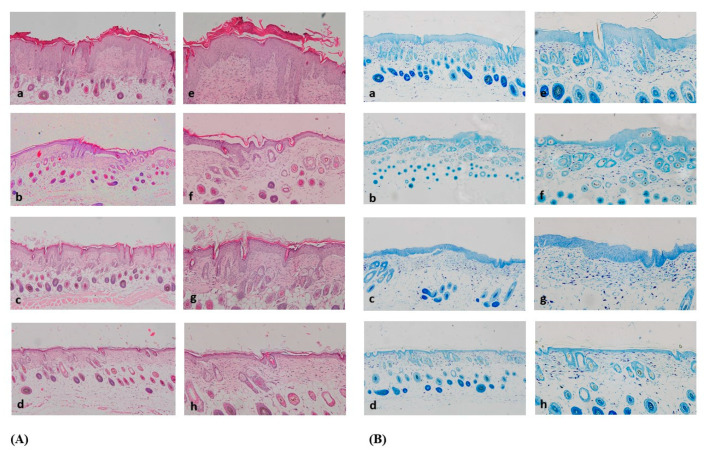
(**A**) H&E staining. Inflammatory cells were exhibited excessively in a, e. These patterns were decreased in b, c, f, g. In d, h, there were few inflammatory cells. (**B**) TB staining. The mast cells were prominent in a, e. These patterns were decreased in b, c, f, g. In d, h, there was an almost normal pattern.; figure a, e: positive control group, figure b, f: Synbio-glucan topical treatment group, figure c, g: Synbio-glucan dietary treatment group, figure d, h: Synbio-glucan topical + dietary treatment group.

**Table 1 nutrients-13-01090-t001:** Serum cytokine antibody array representing the significantly downregulated proteins in the Synbio-glucan topical treatment group compared to the positive control group based on the fold-change rank.

Rank	Antibody Name	Fold-Change	Gene Symbol	Swiss-Prot Entry
Downregulated				
1	IFN-beta	0.354	Ifnb1	P01575
2	GDF-8	0.377	Mstn	O08689
3	Common gamma Chain/IL-2 R gamma	0.408	Il2rg	P34902
4	Endostatin	0.412	Col18a1	P39061
5	IGFBP-3	0.423	Igfbp3	P47878
6	SPARC	0.424	Sparc	P07214
7	WISP-1/CCN4	0.432	Wisp1	O54775
8	TLR2	0.440	Tlr2	Q9QUN7
9	SLPI	0.450	Slpi	P97430
10	MIP2	0.453	Cxcl2	P10889
11	VEGF-B	0.456	Vegfb	P49766
12	CCL28	0.458	Ccl28	Q9JIL2
13	ICAM-1	0.460	Icam1	P13597
14	Fas/TNFRSF6	0.461	Fas	P25446
15	CXCR6	0.465	Cxcr6	Q9EQ16
16	IL-1 RI	0.475	Il1r1	P13504
17	IGFBP-1	0.475	Igfbp1	P47876
18	b FGF	0.482	Fgf2	P15655
19	Prolactin	0.483	Prl	P06879
20	M-CSF	0.485	Csf1	P07141
21	TGF-beta RII	0.485	Tgfbr2	Q62312
22	CRP	0.488	Crp	P14847
23	Lymphotoxin beta R/TNFRSF3	0.488	Ltbr	P50284
24	Frizzled-6	0.496	Fzd6	Q61089
25	IL-27	0.499	Il27	Q8K3I6
26	IL-23 R	0.499	Il23r	Q8K4B4
27	TCCR/WSX-1	0.499	Il27ra	O70394

**Table 2 nutrients-13-01090-t002:** Serum cytokine antibody array representing the significantly upregulated proteins in the Synbio-glucan dietary treatment group compared to the positive control group based on the fold-change rank.

Rank	Antibody Name	Fold-Change	Gene Symbol	Swiss-Prot Entry
Upregulated				
1	IL-9	83.568	Il9	P15247
2	Dtk	62.380	Tyro3	P55144
3	FGF R3	24.767	Fgfr3	Q61851
4	GFR alpha-4/GDNF R alpha-4	16.304	Gfra4	Q9JJT2
5	Thymus Chemokine-1	14.052	Ppbp,	Q9EQI5
6	TRAIL/TNFSF10	9.431	Tnfsf10	P50592
7	Follistatin-like 1	8.738	Fstl1	Q62356
8	VE-Cadherin	6.714	Cdh5	P55284
9	BLC	6.328	Cxcl13	O55038
10	ICAM-2/CD102	5.753	Icam2	P35330
11	IL-22	5.632	Il22	Q9JJY9
12	IL-10 R alpha	5.608	Il10ra	Q61727
13	WIF-1	4.277	Wif1	Q9WUA1
14	MIP-3 beta	3.753	Ccl19	O70460
15	MIP-1alpha	3.608	Ccl3	P10855
16	LIF	3.496	Lif	P09056
17	VEGF-D	3.197	Figf	P97946
18	RANTES	3.117	Ccl5	P30882
19	Decorin	2.988	Dcn	P28654
20	P-Selectin	2.687	Selp	Q01102
21	IL-13	2.309	Il13	P20109
22	IL-1 Ra	2.288	Il1r1	P13504
23	IL-1 R4/ST2	2.245	Il1rl1	P14719
24	PDGF-C	2.180	Pdgfc	Q8CI19
25	CD27 Ligand/TNFSF7	2.173	Cd70	O55237
26	ICK	2.153	Ick	Q9JKV2
27	SDF-1	2.074	Cxcl12	P40224

**Table 3 nutrients-13-01090-t003:** Serum cytokine antibody array representing the significantly up- or downregulated proteins in the Synbio-glucan topical + dietary treatment group compared to the positive control group based on the fold-change rank.

Rank	Antibody Name	Fold-Change	Gene Symbol	Swiss-Prot Entry
Upregulated				
1	Dtk	95.560	Tyro3	P55144
2	IL-9	80.615	Il9	P15247
3	GFR alpha-4/GDNF R alpha-4	38.326	Gfra4	Q9JJT2
4	FGF R3	22.346	Fgfr3	Q61851
5	Follistatin-like 1	16.970	Fstl1	Q62356
6	Thymus Chemokine-1	14.916	Ppbp,	Q9EQI5
7	ICAM-2/CD102	11.732	Icam2	P35330
8	VE-Cadherin	9.709	Cdh5	P55284
9	TRAIL/TNFSF10	8.673	Tnfsf10	P50592
10	Decorin	8.051	Dcn	P28654
11	IL-1 Ra	6.894	Il1r1	P13504
12	ICK	3.760	Ick	Q9JKV2
13	Frizzled-7	3.344	Fzd7	Q61090
14	GDF-5	3.263	Gdf5	P43027
15	IL-1 R4/ST2	2.961	Il1rl1	P14719
16	Common gamma Chain/IL-2 R gamma	2.836	Il2rg	P34902
17	CXCR6	2.769	Cxcr6	Q9EQ16
18	Lungkine	2.718	Cxcl15	Q9WVL7
19	VEGFC	2.509	Vegfc	P97953
20	Glut2	2.441	Slc2a2	P14246
21	Endostatin	2.422	Col18a1	P39061
22	RANTES	2.364	Ccl5	P30882
23	CTACK	2.329	Ccl27	Q9Z1X0
24	LIF	2.310	Lif	P09056
25	IL-28/IFN-lambda	2.298	Il28b	Q8CGK6
26	TCA-3	2.284	Ccl1	P10146
27	IGFBP-2	2.246	Igfbp2	P47877
28	IL-17 R	2.232	Il17ra	Q60943
29	Eotaxin-2	2.216	Ccl24	Q9JKC0
30	IL-31	2.192	Il31	Q6EAL8
31	BLC	2.174	Cxcl13	O55038
32	IL-11	2.150	Il11	P47873
33	HVEM/TNFRSF14	2.145	Tnfrsf14	NP_849262
34	CCL28	2.135	Ccl28	Q9JIL2
35	CRP	2.127	Crp	P14847
36	FLRG(Follistatin)	2.105	Fstl3	Q9EQC7
37	beta-Catenin	2.103	Ctnnb1	Q02248
38	Soggy-1	2.091	Dkkl1	Q9QZL9
39	GDF-8	2.064	Mstn	O08689
40	IGFBP-5	2.040	Igfbp5	Q07079
41	LIX	2.017	Cxcl5	P50228
42	Frizzled-6	2.015	Fzd6	Q61089
43	b FGF	2.011	Fgf2	P15655
44	IFN-beta	2.007	Ifnb1	P01575
45	CCL1/I-309/TCA-3	2.001	Ccl1	P10146
Downregulated				
1	Activin A	0.373	Inhba	Q04998
2	Gremlin	0.403	Grem1	O70326
3	IL-4	0.408	Il4	P07750
4	SLPI	0.416	Slpi	P97430
5	Angiopoietin-like 2	0.425	Angptl2	Q9R045
6	Frizzled-1	0.446	Fzd1	O70421
7	Growth Hormone R	0.447	Ghr	P16882
8	IL-6 R	0.456	Il6ra	P22272
9	ICAM-5	0.469	Icam5	Q60625
10	IL-1 Rb	0.473	Il1r2	P27931
11	Axl	0.475	Axl	Q00993
12	Flt-3 Ligand	0.487	Flt3l	P49772

## Data Availability

We exclude this statement.

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
