# Peer review of "Anti-Inflammatory and Immune Modulatory Effects of Synbio-Glucan in an Atopic Dermatitis Mouse Model"

_nutrients, 2021, doi:10.3390/nu13041090_

Round 1

Reviewer 1 Report

The paper is a very interesting study on the use of beta-glucan on mouse models with AD. The results   showed that topical beta-glucan alleviated skin lesions, alone or combined with  oral administration of β-glucan and probiotics, while oral administration of beta-glucan alone was not statistically significantly associated with a better outcome. Here some queries:

I think you should better assess how you scored the skin lesions...was there a single researcher that did the scoring? Or multiple researchers? I think that a picture for each score grade ( 0 (none), 1 (mild), 2 (moderate), and 3 (severe). ) if available, would be a nice addition to the paper.

I would expand the introduction and talk more about  the clinical manifestations of AD and its available treatments; here some studies I suggest you to check in order to expand the introduction: Clinical characteristics of AD (doi:10.18176/jiaci.0519.) and available treatments of AD (doi:10.1111/dth.12787.)

Thank you

Author Response

The paper is a very interesting study on the use of beta-glucan on mouse models with AD. The results showed that topical beta-glucan alleviated skin lesions, alone or combined with oral administration of β-glucan and probiotics, while oral administration of beta-glucan alone was not statistically significantly associated with a better outcome. Here some queries:

Point 1: I think you should better assess how you scored the skin lesions. Was there a single researcher that did the scoring? Or multiple researchers? I think that a picture for each score grade (0 (none), 1 (mild), 2 (moderate), and 3 (severe)) if available, would be a nice addition to the paper.

Response: After Suto. group (1999) established the score grade, multiple researchers have been used this score grade. We added one more reference (Takakura et al., 2005) in the M&M part. (Line142) And also we added ‘Supplementary Figure’ as your comment (Supplement figure 1) (Line 142). Usually, the scoring is a little subjective, so we performed the blindness assessment.

Point 2: I would expand the introduction and talk more about the clinical manifestations of AD and its available treatments; here some studies I suggest you to check in order to expand the introduction: Clinical characteristics of AD (doi:10.18176/jiaci.0519.) and available treatments of AD (doi:10.1111/dth.12787.)

Response: As your comment, we added the clinical manifestations of AD and its available treatments based on Clinical characteristics of AD (doi:10.18176/jiaci.0519.) and available treatments of AD (doi:10.1111/dth.12787.). (Line 36-40, 45-50) We really appreciate your thoughtful comment.

Thank you so much for your thoughtful advice. We hope this revised manuscript and letter meet your expectation.

Reviewer 2 Report

In their manuscript Kim et al. describ ethe application of what the call Synbio-glucan to treat experimental atopic dermatitis. The present interesting effects. but the present form need to be improved. 

  1. The authors should use a style to present their data that is common in the scientific community. For instance, they start to results section without explaining what they did. From the M&M section or simply by Intuition, it is possible to conclude what was done but. However, this is not acceptable for a scientific manuscript.
  2.  The English language and style needs serious improvement. Often it is not clear what was compared because the style used is ambiguous. 

More specific:

3. Mention in the Abstract that oats were probiotic fermented.

4. M&M: mentioned that NC/NGA mice are an atopic model.

5. M&M: Mention the housing conditions of the mice i.e. SPF or conventional.

6. M&M: HDM is an abreviation. Spell it out first time it occurs in the text. Having done so in the Abstract is not sufficient. 

7. M&M: topical application? was that done by just adding the fluid to the Skin or was some cover added?

8. M&M: section 2.4 what means: four weeks from the start of the experiment? does it mean one week after stopping treatment with HDM? please specify. 

9. M&M: section 2.6. the name cytokine antibody assay does not make sense. Serum cytokine array is meant. 

10. Results: start with description of the Experiment including the time when the samples were taken.

11. Results: line 154: what does mean: antibody array scatter plots. what was done?

12. Figure 2: the lettering is far too small. It is not possible to read the denominations. in the legend more explanations should be given. 

13. Results: line 167-174. To me the sentences are ambiguous. IT is not clear to me, what was compared and how.

14. Tables: from the head lines it is not clear what was compared.

15. Results: line 181: on the Arrays Proteins are identified not genes. 

16. Results: line 181 and follows: only a list of functions is concluded from These data. This is very unsatisfactorial. 

17. Figure 3A: the quality of the pictures is not acceptable. Nothing can be recognized. Enlargements of the affected Areas should b shown and some help e.g. in form of arrows should be provided.

18. Results: Figure 3 and 4. is the difference of the histological score between control and dietary alone significant. Similarly, is the difference beween the topiclal and the topical plus dietary Group significant in figure 3A. the histology Show clear differences between the Groups. Why is this not mentioned. It looks almost like an controversy.  

19: Discussion: the effect of dietary treatment seen in histology is not satisfactorially discussed nor is the discrepancy with the other ayssays explained.

20. Discussion: although the array data represent a large part of the data, they are not satisfactorially discussed and no conclusion is found based on these data. 

Author Response

In their manuscript Kim et al. describe the application of what the call Synbio-glucan to treat experimental atopic dermatitis. The present interesting effects. but the present form need to be improved. 

Point 1: The authors should use a style to present their data that is common in the scientific community. For instance, they start to results section without explaining what they did. From the M&M section or simply by Intuition, it is possible to conclude what was done but. However, this is not acceptable for a scientific manuscript.

Response: Sorry about the condition of this paper. We revised the results part generally. Especially, we corrected this result part base on your more specific comments. We hope this manuscript will meet the requirements.  

Point 2: The English language and style needs serious improvement. Often it is not clear what was compared because the style used is ambiguous. 

Response: Sorry about the poor English language and style. We got a proofread in English correction this time and corrected generally this manuscript.

More specific:

Point 3: Mention in the Abstract that oats were probiotic fermented.

Response: As your comment, we changed probiotics to fermented probiotics in the abstract part. (Line 17)

Point 4: M&M: mentioned that Nc/Nga mice are an atopic model.

Response: As your comment, we mention that Nc/Nga mice are an atopic model in the M&M part. (Line 77)

Point 5: M&M: Mention the housing conditions of the mice i.e. SPF or conventional.

Response: We added the housing condition as conventional in the M&M part. (Line 78). Thank you for your comment.

Point 6: M&M: HDM is an abbreviation. Spell it out first time it occurs in the text. Having done so in the Abstract is not sufficient. 

Response: We spelled it out first time in the M&M part. (Line 83) Sorry about the missing.

Point 7: M&M: topical application? was that done by just adding the fluid to the skin or was some cover added?

Response: We applied fluid to the skin and wait to permeate to the skin. After drying Synbio-glucan topical treatment, all mice were replaced to their inhabited cases. We mentioned this in the M&M part. (Line 113-114)

Point 8: M&M: section 2.4 what means: four weeks from the start of the experiment? does it mean one week after stopping treatment with HDM? please specify. 

Response: With this protocol, the period of treatment of Syn-bio glucan and treatment with HDM was overlapped. We made and added the experimental scheme as figure 1. (Line 100)

Point 9: M&M: section 2.6. the name cytokine antibody assay does not make sense. Serum cytokine array is meant. 

Response: RayBiotech Inc. provided this array name as “Mouse Cytokine Antibody Array”. So we used the name “cytokine antibody assay”. Please understand about this.

Point 10: Results: start with description of the Experiment including the time when the samples were taken.

Response: Sorry about the missing of description of the experiment. The serum, which was used in this study, was collected from sacrificed mice. We added the description of experiment in the text. (Line 166-168, 186-188)

Point 11: Results: line 154: what does mean: antibody array scatter plots. what was done?

Response: With scatter plots, we could recognize up- or down- changes between the treatment groups and the positive control group at a glance. We suggested this plot to show the patterns of up- or down-  changes between the treatment groups and the positive control group. We added this description in the text. (Line 186-188, 192-195)

Point 12: Figure 2: the lettering is far too small. It is not possible to read the denominations. in the legend more explanations should be given. 

Response: Sorry about the impropriated lettering of the figures. Now we enlarged the lettering of the figures. And We also changed the functional analysis graph (B, C, D) for more proper understanding. If more enlargement is need, we can provide figure 2 (Now it became figure 3) as another jpeg files. And also we described more in the legend part of figure 2 (Now figure 3). (Line 198-212)

Point 13: Results: line 167-174. To me the sentences are ambiguous. It is not clear to me, what was compared and how.

Response: Sorry about the ambiguous expression. All of data in the treatment groups were compared with the data in the positive control group. We tried to clarify what was compared and how in the manuscript. (Line 214-224) We hope this is satisfactory to the reviewer.

Point 14: Tables: from the head line, it is not clear what was compared.

Response: All of data in the treatment groups were compared with the data in the positive control group. We corrected the head line to clear these, as your comment. (Line 225-228, 229-232, 233-236)

Point 15: Results: line 181: on the Arrays Proteins are identified not genes. 

Response: Sorry about the mistake. We corrected ‘genes’ to ‘proteins’. (Line 237)

Point 16: Results: line 181 and follows: only a list of functions is concluded from these data. This is very unsatisfactory. 

Response: Sorry about the deficiency. We added the information about GO and KEGG in the manuscript. And we discussed more about the results in the discussion part. (Line 240-241, 390-397)

Point 17: Figure 3A: the quality of the pictures is not acceptable. Nothing can be recognized. Enlargements of the affected areas should be shown and some help e.g. in form of arrows should be provided.

Response: We enlarged the pictures and affected areas were indicated this time (Figure4A). (Line 287)

Point 18: Results: Figure 3 and 4. is the difference of the histological score between control and dietary alone significant. Similarly, is the difference between the topical and the topical plus dietary group significant in figure 3A. the histology Show clear differences between the Groups. Why is this not mentioned. It looks almost like a controversy.  

Response: In this study, we could not confirm the enough improvement in Synbio-glucan dietary treatment group. We could find the most profound effect in Synbio-glucan topical + dietary treatment group. The second improvement effect could be confirmed in Synbio-glucan topical treatment group We revised the description in the result part to avoid the confusion. (Line 303-307, 313-315) Sorry about this confusion.

Point 19: Discussion: the effect of dietary treatment seen in histology is not satisfactory discussed nor is the discrepancy with the other assays explained.

Response: The scores of skin lesions and histological analysis in this study showed that the Synbio-glucan topical + dietary treatment markedly alleviated the skin lesions. Both the Synbio-glucan topical treatment and the Syn-bio-glucan dietary treatment were also alleviated the skin lesions, however, the improvement of skin legion in the Synbio-glucan dietary treatment was very lower. This is the reason why we could not confirm and discuss about the profound improvement in the Syn-bio-glucan dietary treatment. We presumed the high percentage of β-glucan derived from oats dietary treatment can show the improvement in the skin lesion. We described about this in the discussion part. (Line 350-367)

Point 20: Discussion: although the array data represent a large part of the data, they are not satisfactory discussed and no conclusion is found based on these data. 

Response: As your comment, we added the discussion about array data. And also we discussed about each interesting cytokine, especially IFN-β, IL-9, in the discussion part (Line 373-397). 

Thank you so much for your thoughtful advice. We hope this revised manuscript and letter meet your expectation.

Round 2

Reviewer 1 Report

The authors responded to all queries. No more comments.

Author Response

The authors responded to all queries. No more comments.

Response: We really appreciate all of your thoughtful advice. 

Reviewer 2 Report

The manuscript by Kim et al. has very much improved. Two minor points should possibly be addressed:

  1. line 77 and 325: "mouse model" instead of mice model.
  2. the beginning of the results is still not appropriate. I suggest to start like follows: Atopic dermatitis was induced in  Nc/Nga mice. Such mice were treated with Symbio-glucan topically, dietary, dietary plus topically or left untreated. Normally, atopic.....

Author Response

The manuscript by Kim et al. has very much improved. Two minor points should possibly be addressed.

Point 1. line 77 and 325: "mouse model" instead of mice model.

Response: As your comment, we corrected "mouse model" instead of mice model. (Line 18, 68, 292, 350, 357, 358)

Point 2. the beginning of the results is still not appropriate. I suggest to start like follows: Atopic dermatitis was induced in Nc/Nga mice. Such mice were treated with Symbio-glucan topically, dietary, dietary plus topically or left untreated. Normally, atopic.....

Response: As your comment, we added this description in Line 146-147. We really appreciate your kindly advice.